# The Association Between Decompensated Liver Cirrhosis and Deep Neck Infection: Real-World Evidence

**DOI:** 10.3390/ijerph16203863

**Published:** 2019-10-12

**Authors:** Ming-Shao Tsai, Geng-He Chang, Wei-Ming Chen, Chia-Yen Liu, Meng-Hung Lin, Pey-Jium Chang, Tsung-Yu Huang, Yao-Te Tsai, Ching-Yuan Wu, Cheng-Ming Hsu, Yao-Hsu Yang

**Affiliations:** 1Department of Otolaryngology—Head and Neck Surgery, Chiayi Chang Gung Memorial Hospital, Chiayi 613, Taiwan; b87401061@cgmh.org.tw (M.-S.T.);; 2Graduate Institute of Clinical Medical Sciences, College of Medicine, Chang Gung University, Taoyuan 33302, Taiwan; 3Health Information and Epidemiology Laboratory, Chiayi Chang Gung Memorial Hospital, Chiayi 613, Taiwan; 4Division of Gastroenterology and Hepatology, Department of Internal Medicine, Chiayi Chang Gung Memorial Hospital, Chiayi 613, Taiwan; 5Division of Infectious Diseases, Department of Internal Medicine, Chiayi Chang Gung Memorial Hospital, Chiayi 613, Taiwan; 6Department of Traditional Chinese Medicine, Chiayi Chang Gung Memorial Hospital, Chiayi 613, Taiwan; 7School of Traditional Chinese Medicine, College of Medicine, Chang Gung University, Taoyuan 33302, Taiwan

**Keywords:** liver, hepatic, cellulitis, abscess, comorbidities, risk factor

## Abstract

Background: Deep neck infection (DNI) can progress to become a life-threatening complication. Liver cirrhosis, which is related to poor immune conditions, is a likely risk factor for DNI. This study investigated the risk and mortality of DNI in patients with decompensated liver cirrhosis (DLC). Methods: We performed a nationwide cohort study using the National Health Insurance Research Database (NHIRD) in Taiwan. We included a total of 33,175 patients with DLC between 2000 and 2013, from the Catastrophic Illness Patient Database, a subsection of the NHIRD, along with 33,175 patients without cirrhosis who were matched in a 1:1 proportion for age, sex, and socioeconomic status. The occurrence of DNI was the primary study outcome. The risk, treatment, and mortalities of DNI were evaluated in the study and comparison cohorts. Results: DLC Patients had a significantly higher incidence of DNI than noncirrhotic patients (*p* < 0.001). The adjusted Cox proportional hazard regression showed that DLC was associated with a significantly higher risk of DNI (adjusted hazard ratio, 4.11; 95% confidence interval, 3.16–5.35, *p* < 0.001). The mortality rate in cirrhotic patients with DNI was not significantly higher than that in noncirrhotic patients with DNI (11.6% vs. 9.8%; *p* = 0.651). Conclusions: This study is the first to investigate the correlation between DLC and DNI. The study findings strongly indicate that DLC is an independent risk factor for DNI. Cirrhotic patients with DNI do not have a significantly poorer survival rate than noncirrhotic patients with DNI. Therefore, physicians should be alert to potential DNI occurrence in DLC patients. Besides this, intensive care and appropriate surgical drainage can yield similar survival outcomes in DLC-DNI and noncirrhosis-DNI patients.

## 1. Introduction

Deep neck infection (DNI) occurs in cervical fascial spaces. The incidence rate of DNI is 20.88 per 100 thousand persons; years are reported in a recent population-based study in Taiwan [1]. The incidence of DNI has declined with the widespread availability of antibiotics; however, it can progress to become a life-threatening complication [2]. The sources of DNI arise from infections of the teeth, tonsils, lymph nodes, salivary glands, or malignancies, and subsequently progress to abscesses of the deep neck spaces [3]. These spaces are connected to the mediastinum, carotid sheath, skull base, and meninges. After the infection reaches these important areas, mortality can occur [4]. After abscess formation, surgery is considered the mainstay of treatment; however, conservative medical treatments are effective in selective cases [2]. Previous studies have demonstrated that the predisposing risk factors for DNI include diabetes mellitus (DM), end-stage renal disease (ESRD), drug abuse, congenital neck cysts, aging, and poor oral hygiene [5,6,7,8,9,10,11]. Viera et al. suggested that immunosuppression due to other sources, such as hepatic disease, HIV infection, and chemotherapy could be risk factors [5]. Previous studies have revealed that cirrhosis is a common underlying disease in patients with DNI [6,12,13,14,15,16,17]. DNI are often polymicrobial, and the most predominant pathogen is Viridans Streptococci [18]. Furthermore, the viridans streptococci has been reported as a common species of infection in DLC patients [19].

According to our review of the relevant literature, despite some case reports, the incidence and mortality of DNI among patients with decompensated or compensated cirrhosis remained unknown [6,12,13,14,15,16,17]. Therefore, this study investigated the risk, treatment modalities, and mortalities of DNI in patients with DLC.

## 2. Methods

### 2.1. Data Sources

We performed a nationwide cohort study using population-based data from the National Health Insurance Research Database (NHIRD) in Taiwan. The National Health Insurance (NHI) program is a single-payer and compulsory health insurance that, at present, covers all forms of healthcare services for 99.6% of the population in Taiwan [20,21,22]. The high coverage rate empowers the use of NHIRD data for conducting nationwide and population-based studies [22]. The NHIRD contains extensive information, including prescription drugs, clinical visits, surgical procedures, and diagnostic codes [20,23,24]. In the NHIRD, the International Classification of Diseases, Ninth Revision, Clinical Modification (ICD-9-CM), and the Procedure Coding System (ICD-9-PCS) are adopted to define the diagnostic and procedure codes, respectively [25,26]. The NHIRD has been used for various studies and provides high quality information on diagnoses, hospitalizations, and prescriptions [27,28].

The Institutional Review Board of Chang Gung Memorial Hospital approved this study (CGMH-IRB No.201601249B1). Following strict confidentiality guidelines in accordance with the personal electronic data protection regulations, the personal information of all patients in the NHIRD was encrypted. 

### 2.2. Study Cohort

Figure 1 presents the flowchart of the patient enrollment. The study cohort was identified from the NHIRD. DLC was defined on the basis of the specific admission codes (ICD-9-CM 571.2, 571.5, and 571.6) and the certification from the Registry for Catastrophic Illness Patient Database (RCIPD), a subsection of the NHIRD that contains all claim data for 2000–2013 [29,30]. Patients with DLC must satisfy at least one of the following criteria to be registered in the RCIPD: (1) Hepatic coma or liver decompensation, (2) variceal bleeding, or (3) intractable ascites [31]. The definition of DLC in this study is compatible with the European Association for the Study of the Liver (EASL) clinical practice guideline [32]. Patients who had DNI before receiving a diagnosis of DLC were excluded from the study.

### 2.3. Comparison Cohort

We matched each patient with DLC with one comparison patient without cirrhosis selected from the Longitudinal Health Insurance Database in 2000 (LHID2000), a representative database of 1 million random patients from the 2000 registry of all NHI enrollees, which contained all claim data for 1996–2013 [21,30,33]. According to Taiwan’s National Health Research Institutes Reports, the sample group and all enrollees in the LHID2000 did not differ significantly in their age, sex, or health-care costs [21,30,33]. The comparison patients were matched by age, sex, socioeconomic status, and index date. The date of the first registry was considered the index date for patients with DLC. The index date for the comparison patients was obtained by matching the cirrhotic patient’s index date. The comparison patients, diagnosed as having DNI before enrollment, were excluded from the study.

### 2.4. Outcome and Covariates

Patients were followed until death or the end of the study period (31 December 2013) [34,35]. The primary study outcome was the occurrence of DNI (inpatient settings coded as ICD-9-CM 478.22 [parapharyngeal abscess], 478.24 [retropharyngeal abscess], 528.3 [cellulitis and abscess of the oral soft tissue], and 682.11 [cellulitis and abscess of the neck]).

For patients who finally developed DNI, the treatment modalities, hospitalization duration, tracheostomy, intensive care unit (ICU) care, mediastinitis (ICD-9-CM codes: 510, 513, and 519.2), and mortalities were investigated. The patients who underwent surgical drainage were categorized into the “surgery group”, and those who received abscess aspiration or antibiotics without surgery were categorized into the “non-surgery group”.

Patient sociodemographic information, including age, sex, socioeconomic status, and urbanicity level, was obtained from enrollment data files. The comorbidities of patients, including DM, chronic kidney disease (CKD), and autoimmune disease, were retrieved from the ambulatory and inpatient claims data. We included these comorbidities if they were observed either in inpatient settings or in three or more outpatient claims. Each comorbidity was analyzed as a binominal variable.

### 2.5. Statistical Analyses

Continuous variables and descriptive statistics were analyzed using the independent student t test and Pearson chi-squared test, respectively, to compare the cirrhosis and noncirrhosis groups in terms of their sociodemographic characteristics and comorbidities. The Kaplan–Meier method was used to calculate the cumulative incidences of DNI, and the log-rank test was applied to compare the differences between each curve. The adjusted hazard ratio (HR) for DNI was estimated using the Cox proportional hazard regression models. SAS (version 9.4, SAS Inc., Cary, NC, USA) was used for statistical analysis, and a two-sided p value of <0.05 was considered statistically significant.

## 3. Results

### 3.1. Clinical Characteristics of Cirrhosis and Noncirrhosis Groups

The present study included 33,175 patients with DLC and 33,175 comparison patients without cirrhosis between 2000 and 2013 in Taiwan. After matching for sex, age, and socioeconomic status, the results revealed that the cirrhosis group was more likely to develop autoimmune disease, DM, and ESRD than the noncirrhosis group (Table 1). The mean (standard deviation) observation period in this study was 5.8 (3.9) years for the cirrhosis cohorts and 9.2 (3.1) years for the noncirrhosis cohorts. During the follow-up period, 23,066 DLC patients and 1957 noncirrhosis patients died, respectively.

### 3.2. Cumulative Incidence of DNI in the Study and Comparison Cohorts

Using the Kaplan–Meier method, the cumulative incidence of DNI was observed to be significantly higher in the cirrhosis group than in the noncirrhosis group (*p* < 0.001; Figure 2). The 4, 8, and 12-year death-adjusted cumulative incidences of DNI were 0.435% versus 0.103%, 0.946% versus 0.216%, and 1.651% versus 0.326%, respectively, in the study and comparison cohorts.

The Kaplan–Meier analysis demonstrated the cumulative DNI identified in the study and comparison cohorts, respectively, during the follow-up period (1997–2013). The log-rank test revealed a significantly higher cumulative incidence in the DLC cohort (*p* < 0.001). The ‘No. at risk’ is the number of patients who were still alive and whose follow-up extended at least that far into the curve.

### 3.3. Decompensated Cirrhosis as an Independent Risk Factor for DNI

Multivariate analysis was conducted to compare the risk of DNI in the study and comparison cohorts across different sexes, age groups, urbanicity levels, socioeconomic statuses, and comorbidities (Table 2). No significant differences were observed across different sexes, age groups, urbanicity levels, income levels, or comorbidities, except for DLC and diabetes mellitus. Patients with DLC were at a significantly higher risk of DNI than those without cirrhosis, after adjusting for other variables (HR, 4.11; 95% CI, 3.16–5.35; *p* < 0.001).

### 3.4. Treatment Modalities and Mortalities

We analyzed the treatment modalities and mortalities of those patients in the study and comparison cohorts who finally developed DNI. The ratios of patients receiving surgical drainage, tracheostomy, mean hospitalization duration, mediastinitis, and three-month mortality did not differ significantly between the DLC-DNI and noncirrhosis-DNI groups (Table 3). The ratio of ICU care in DNI patients with DLC was higher than in those without cirrhosis. The three-month mortality rates of DNI were 11.6% versus 9.8% (*p* = 0.651), respectively, in the DLC-DNI and noncirrhosis-DNI groups.

## 4. Discussion

According to our review of the relevant literature, this study is the first to investigate the risk and mortality of DNI in patients with liver cirrhosis. Our study showed that patients with DLC were at a significantly increased DNI risk than those without cirrhosis. In our study, the 4, 8, and 12-year death-adjusted cumulative incidences of DNI in the cirrhosis group were 0.435%, 0.946%, and 1.651%, respectively. This study is the first to estimate these indices. After adjusting for the potential confounders, the HR for DNI was 4.11 (95% CI, 3.16–5.35), indicating that the risk of DNI increased by 311% in patients with DLC. The log-rank test results indicated that patients with DLC had a significantly higher incidence of DNI than those without cirrhosis (*p* < 0.001). In our study, the mean (standard deviation) follow-up time was 5.8 (3.9) and 9.2 (3.1) years in the study and comparison cohorts, respectively. This observation duration was adequate to assess the risk of DNI among each group. Furthermore, the treatment modalities did not differ significantly among cirrhotic and noncirrhotic patients with DNI. The three-month mortality rates of DNI in the cirrhosis group were not significantly higher than in the non-cirrhosis group.

Previous studies reported that patients with DLC had a five-fold higher incidence of bacterial infections than the general population [36,37,38]. Liver disease has been reported to be associated with a higher risk of odontogenic infection, which is a common source of DNI [1,39]. Systemic inflammation response syndrome or sepsis occurs during hospitalization or is present at admission in approximately 30% of patients with DLC [40,41]. Spontaneous bacterial peritonitis (SBP), urinary tract infections, pneumonia, skin and soft tissue infections, and bacteremia are common etiologies of sepsis. These findings support our observation that DLC is associated with DNI. The severity of infection was higher in patients with DLC, who are more likely to die due to sepsis than those without cirrhosis. Studies have reported that progressive liver failure, variceal bleeding, SBP history, and hospitalization are associated with an increased risk of bacterial infection [36,37,38]. Bacterial infections result in a 3.75-fold increase in mortality in patients with DLC, reaching 30% at one-month and 63% at one-year [42,43]. In our study, even though the DLC-DNI group had a higher ratio of ICU care, there was no difference in the mortality rate in DLC-DNI patients and that in noncirrhosis-DNI. This finding implied that intensive care and appropriate surgical drainage could yield similar survival outcomes in DLC-DNI and noncirrhosis-DNI patients.

Cirrhosis-associated immune dysfunction refers to immunodeficiency and systemic inflammation that occur during cirrhosis [44]. Cirrhosis can impair the intestinal–portal route barrier, thus facilitating bacterial entry into the systemic circulation and increasing the susceptibility to various infectious diseases, such as respiratory infections, spontaneous bacterial peritonitis, or urinary tract infections [45,46]. The concomitant presence of impaired phagocytosis, poor opsonization capacity, low complement levels, and impaired reticuloendothelial system in patients with cirrhosis increases their susceptibility to bacteremia [42,45,47]. Therefore, patients with cirrhosis can easily develop sepsis, including DNI.

The adjusted HR for DM in the present study was 1.81 (CI, 1.44–2.29, *p* < 0.001), which is consistent with the findings of previous studies [5,6,7,8]. However, the adjusted HR for autoimmune disease and ESRD was nonsignificant after adjusting for sex, age, urbanicity levels, income levels, and comorbidities because the present study evaluated the effects of DLC on the risk of DNI, and therefore the study patients were different from the general population.

As cirrhosis and DNI are not very common diseases, it is difficult for a single hospital to collect a sufficiently large sample, with a sufficient follow-up time, and to specifically investigate the correlation between cirrhosis and DNI. Using the nationwide population-based data from the NHIRD, we identified all cases of DLC in Taiwan with minimal selection bias. We ensured the diagnostic accuracy of DLC by using the RCIPD and ascertained the occurrence of DNI by investigating only hospitalized patients whose diagnoses were strictly coded for reimbursement purposes. Furthermore, we were able to investigate the treatment modalities, including surgical drainage, hospitalization duration, tracheostomy, ICU care, mediastinitis, and mortalities, because the NHI program covered all healthcare services. To elucidate the association between DLC and the risk of DNI, we excluded patients who had DNI before receiving a diagnosis of DLC. To reduce the potential confounders, we used adjusted HRs (after adjusting for DM, CKD, and autoimmune disease) to compare the outcomes between the study and comparison cohorts.

The present study has some limitations. First, detailed data on albumin, bilirubin, and prothrombin time are not available in the NHIRD. Second, the exact etiology of cirrhosis was not identified in our study, and we were unable to evaluate the severity of cirrhosis based on the Child–Pugh scores. Third, the patients in the study cohort were selected from the RCIPD, which only included patients with DLC. Therefore, patients with compensated liver cirrhosis were not included in this study. Nonetheless, despite the aforementioned limitations, this study is the most comprehensive nationwide population-based study to prove that DLC is a risk factor for DNI. Moreover, additional investigations are warranted to determine the risk of DNI in patients with compensated cirrhosis. Finally, our study is a retrospective population-based study. Therefore, additional prospective studies are required to elucidate the relationship between cirrhosis and DNI.

## 5. Conclusions

This study is the first to demonstrate that patients with DLC are at an increased risk of DNI. In addition, cirrhotic patients with DNI do not have a significantly poorer survival rate than noncirrhotic patients with DNI. Our study findings suggest that physicians should consider the risk of DNI while treating patients with DLC. Besides this, intensive care and appropriate surgical drainage can yield similar survival outcomes in DLC-DNI and noncirrhosis-DNI patients.

## Figures and Tables

**Figure 1 ijerph-16-03863-f001:**
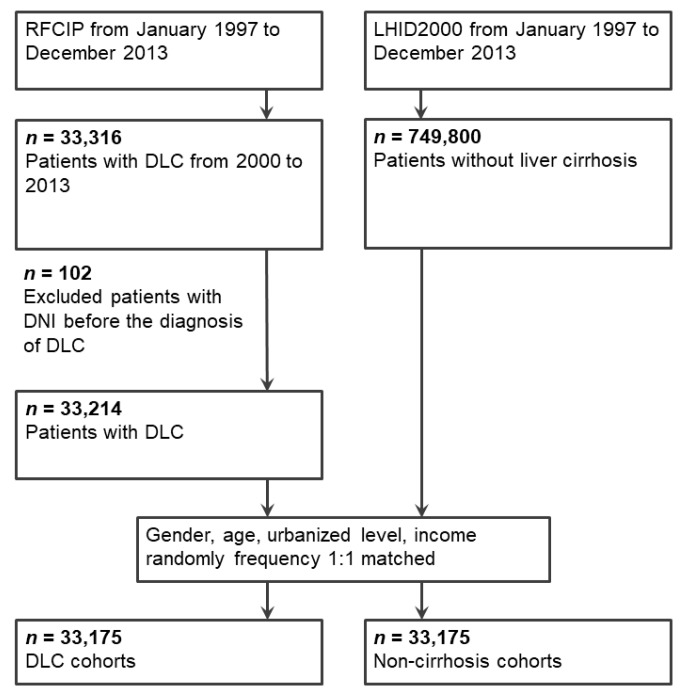
Enrolment schema of the study and matched cohorts. Abbreviations: RFCIP, Registry for Catastrophic Illness Patients; LHID2000, Longitudinal Health Insurance Database 2000; DLC, decompensated liver cirrhosis; DNI, deep neck infection.

**Figure 2 ijerph-16-03863-f002:**
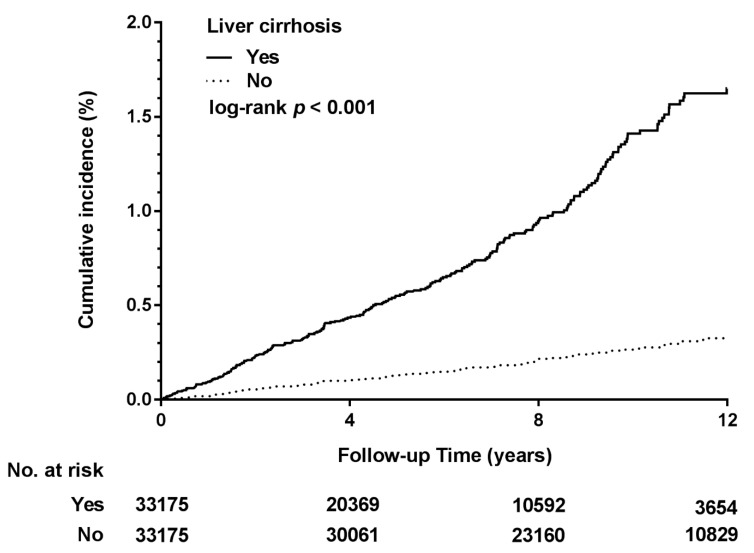
Cumulative incidence of deep neck infection (DNI) for DLC versus noncirrhosis cohorts.

**Table 1 ijerph-16-03863-t001:** Demographic characteristics of the decompensated liver cirrhosis (DLC) and non-cirrhosis cohorts.

Variables	DLC		Non-Cirrhosis		*p*-Value
	(*n* = 33175)		(*n* = 33175)		
	*n*	%	*n*	%	
**Gender**					1.000
Male	24,320	73.3	24,320	73.3	
Female	8855	26.7	8855	26.7	
**Age (years)**					1.000
<40	6851	20.7	6851	20.7	
40–64	20,099	60.6	20,099	60.6	
≥65	6225	18.8	6225	18.8	
**Urbanized level**					1.000
1 (City)	6611	19.9	6611	19.9	
2	14,317	43.2	14,317	43.2	
3	6377	19.2	6377	19.2	
4 (Villages)	5870	17.7	5870	17.7	
**Income (NTD per month)**					1.000
0	4764	14.4	4764	14.4	
1–15,840	7212	21.7	7212	21.7	
15,841–25,000	16,446	49.6	16,446	49.6	
≥25,001	4753	14.3	4753	14.3	
**Covariates**					
Autoimmune disease	887	2.7	610	1.8	<0.001
Diabetes mellitus	12,441	37.5	5316	16.0	<0.001
ESRD	848	2.56	347	1.05	<0.001
**DNI**	242	0.7	82	0.3	<0.001
**Death**					<0.001
Yes	23,066	69.53	1957	5.9	
No	10,109	30.5	31,758	94.1	

Abbreviations: DLC, decompensated liver cirrhosis; ESRD, end-stage renal disease; DNI, deep neck infection; NTD, new Taiwan dollar.

**Table 2 ijerph-16-03863-t002:** Multivariable Cox proportional hazards model for DNI associated with DLC, gender, age, and covariates.

Variables	Crude	95% CI		*p* Value	Adjusted	95% CI		*p* Value
	HR				HR *			
**DLC vs. Non-cirrhosis**								
Non-cirrhosis	1.00				1.00			
DLC	4.91	3.81	6.32	<0.001	4.11	3.16	5.35	<0.001
**Gender**								
Female	1.00				1.00			
Male	1.09	0.85	1.40	0.487	1.08	0.83	1.42	0.591
**Age (years)**								
<40	1.00				1.00			
40–64	1.07	0.82	1.39	0.636	1.03	0.78	1.36	0.838
≥65	0.74	0.50	1.09	0.122	0.78	0.51	1.18	0.230
**Covariates**								
Autoimmune disease	1.04	0.51	2.09	0.919	0.86	0.42	1.73	0.668
Diabetes mellitus	2.63	2.12	3.28	<0.001	1.81	1.44	2.29	<0.001
ESRD	1.61	0.80	3.24	0.186	1.00	0.49	2.02	0.994

Abbreviations: DNI, deep neck infection; DLC, decompensated liver cirrhosis; ESRD, end-stage renal disease; HR, hazard ratio; CI, confidence interval. * The adjusted HR is adjusted for gender, age, urbanized level, income, and comorbidities.

**Table 3 ijerph-16-03863-t003:** Treatment modalities and severity of the DNI patients.

	DLC-DNI		Non-Cirrhosis-DNI		*p* Value
	*n* = 242		*n* = 82		
	*n*	%	*n*	%	
**Treatment modalities ***					0.245
Non-surgery	152	62.8	57	69.5	
Surgery	90	37.2	25	30.5	
**Severity ***					
Tracheostomy	19	7.9	5	6.1	0.600
Hospitalization ^†^ (day)	14.3 ± 15.1		12.5 ± 15.9		0.349
ICU care	46	19.0	7	8.5	0.027
Mediastinitis	9	3.7	1	1.2	0.461
Mortality	28	11.6	8	9.8	0.651

* Fisher exact tests. ^†^ Student’s *t* tests. Abbreviations: DNI, deep neck infection; ICU, intensive care unit; DLC, decompensated liver cirrhosis.

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
