# Peer review of "The Association Between Decompensated Liver Cirrhosis and Deep Neck Infection: Real-World Evidence"

_ijerph, 2019, doi:10.3390/ijerph16203863_

Round 1
Reviewer 1 Report
The authors conducted a nationwide cohort study about Deep neck infection (DNI) and Decompensated liver cirrhosis (DLC) using the National Health Insurance Research Database (NHIRD) of Taiwan. Their results revealed that DLC was an independent risk factor of DNI and that DLC did not affect the mortality of DNI patients through more ICU treatment. Their observation will give physicians who treat DLC patients an enlightenment to pay attention for DNI.
Some point should be reconsidered.
An explanation about No. at risk of Figure 2 should be added in the figure legend.
The last sentence of the paragraph 2.1, "This study was approved..." is already described in the same paragraph as CGMH-IRB No. 201601249B1.
Line 4 from the bottom of page 2, The authors show that patients with DLC must satisfy at least one of the following criteria to be registered in the RCIPD.
hepatic coma or liver decompensation. This will represent DLCs. However, 2. variceal bleeding and 3. intractable ascites can not always represent DLC. Can't DLC group include untrue DLCs? An explanation should be added.
Author Response
Response:
Thank you for your time and consideration in reviewing this manuscript. We have reviewed the English language and spell in the manuscript and have confirmed that the text words presented in the manuscript is without error. We totally agree and thank you for your valuable comments. We have revised the manuscript according to your comments.
Point 1:
An explanation about No. at risk of Figure 2 should be added in the figure legend.
Response to point 1:
Thank you for your valuable comments. We have added an explanation in the legend of figure 2.
Point 2:
The last sentence of the paragraph 2.1, "This study was approved..." is already described in the same paragraph as CGMH-IRB No. 201601249B1.
Response to point 2:
Thank you for your comments. We have removed the duplicated sentence of the paragraph 2.1.
Point 3:
Line 4 from the bottom of page 2, The authors show that patients with DLC must satisfy at least one of the following criteria to be registered in the RCIPD. hepatic coma or liver decompensation. This will represent DLCs. However, 2. variceal bleeding and 3. intractable ascites cannot always represent DLC. Can't DLC group include untrue DLCs? An explanation should be added.
Response to point 3:
The definition of DLC in this study is compatible with the European Association for the Study of the Liver (EASL) clinical practice guideline. DLC is marked by the development of overt clinical signs, the
most frequent of which are ascites, bleeding, encephalopathy, and jaundice. We have rephrased this section and added one reference (Reference 32).
Thank you again for your highly professional comments.
Reviewer 2 Report
The authors investigated the association between deep neck infection and decompensated liver cirrhosis. They found that patients with DLC had higher risk for deep neck infection. Further, the mortality rate was comparable between DLC patients and non-DLC patients although the events associated with DNI was increased in patients with DLC.
The topic of this study is not so novel. Many doctors easily expect the high incidence of severe infection in patients with DLC. However, because of the rarity of DLC and DNI, it is difficult to show the association. The authors conducted a large-scale cohort study using a national database, and demonstrated a clear evidence. Furthermore, the result that similar survival outcome can be obtained with appropriate treatment is very important for clinicians who treat DNI.
Author Response
We are grateful for your generous comments.
Thank you for your time and consideration in reviewing this manuscript.
Reviewer 3 Report
The authors of this manuscript aims to find an association between deep neck infection (DNI) and decompensated liver cirrhosis (DLC). This is the first epidemiological analysis on this topic to my knowledge.
The authors must better introduce the reason to analyze the association between DNI and DLC. In the introduction, the authors should add a brief description of the main microbiological infectors and if there is a relation between this agents and liver.
The authors have to insert epidemiological data, how much is diffused this infection?
Other article have found an association between liver and periodontal disease (PMID 30300961) and despite the literature is scarce on this topic, there is a clinical case that relate this two diseases (PMID 20051062). The authors should discuss these article in the discussion
Author Response
Response:
Thank you for your time and consideration in reviewing this manuscript. We totally agree and thank you for your valuable comments. We have revised the manuscript according to your comments.
Point 1:
The authors must better introduce the reason to analyze the association between DNI and DLC. In the introduction, the authors should add a brief description of the main microbiological infectors and if there is a relation between this agents and liver.
Response to point 1:
Thank you for addressing this important concern. We have revised the introduction according to your comments. Two references have been added (Reference 18 and 19).
Point 2:
The authors have to insert epidemiological data, how much is diffused this infection?
Response to point 2:
Thank you for your valuable comment. We have added the epidemiological data based on your suggestion. One reference has been added (Reference 1).
Point 3:
Other article have found an association between liver and periodontal disease (PMID 30300961) and despite the literature is scarce on this topic, there is a clinical case that relate this two diseases (PMID 20051062). The authors should discuss these articles in the discussion.
Response to point 3:
Yes, we have discussed these important articles in the introduction and discussion sections. Two references have been added (Reference 17 and 39).
Thank you again for your highly professional comments.